# Dietary Long-Chain n-3 Polyunsaturated Fatty Acid Supplementation Alters Electrophysiological Properties in the Nucleus Accumbens and Emotional Behavior in Naïve and Chronically Stressed Mice

**DOI:** 10.3390/ijms23126650

**Published:** 2022-06-14

**Authors:** Mathieu Di Miceli, Maud Martinat, Moïra Rossitto, Agnès Aubert, Shoug Alashmali, Clémentine Bosch-Bouju, Xavier Fioramonti, Corinne Joffre, Richard P. Bazinet, Sophie Layé

**Affiliations:** 1Laboratoire NutriNeuro, UMR INRAE 1286, Bordeaux INP, Université de Bordeaux, 146 Rue Léo Saignat, 33076 Bordeaux, France; m.dimiceli@worc.ac.uk (M.D.M.); maud.martinat@inrae.fr (M.M.); moira.rossitto@inrae.fr (M.R.); agnes.aubert@inrae.fr (A.A.); clementine.bosch-bouju@inrae.fr (C.B.-B.); xavier.fioramonti@inrae.fr (X.F.); corinne.joffre@inrae.fr (C.J.); 2Worcester Biomedical Research Group, School of Science and the Environment, University of Worcester, Worcester WR2 6AJ, UK; 3International Research Network Food4BrainHealth; richard.bazinet@utoronto.ca; 4Department of Clinical Nutrition, Faculty of Applied Medical Sciences, King Abdulaziz University, Jeddah 22254, Saudi Arabia; shoug.alashmali@mail.utoronto.ca; 5Department of Nutritional Sciences, Faculty of Medicine, University of Toronto, Toronto, ON M5S 1A1, Canada

**Keywords:** DHA, EPA, ALA, chronic social defeat stress, lipid microarray, whole-cell patch-clamp electrophysiology, long-term depression, emotional behavior

## Abstract

Long-chain (LC) n-3 polyunsaturated fatty acids (PUFAs) have drawn attention in the field of neuropsychiatric disorders, in particular depression. However, whether dietary supplementation with LC n-3 PUFA protects from the development of mood disorders is still a matter of debate. In the present study, we studied the effect of a two-month exposure to isocaloric diets containing n-3 PUFAs in the form of relatively short-chain (SC) (6% of rapeseed oil, enriched in α-linolenic acid (ALA)) or LC (6% of tuna oil, enriched in eicosapentaenoic acid (EPA) and docosahexaenoic acid (DHA)) PUFAs on behavior and synaptic plasticity of mice submitted or not to a chronic social defeat stress (CSDS), previously reported to alter emotional and social behavior, as well as synaptic plasticity in the nucleus accumbens (NAc). First, fatty acid content and lipid metabolism gene expression were measured in the NAc of mice fed a SC (control) or LC n-3 (supplemented) PUFA diet. Our results indicate that LC n-3 supplementation significantly increased some n-3 PUFAs, while decreasing some n-6 PUFAs. Then, in another cohort, control and n-3 PUFA-supplemented mice were subjected to CSDS, and social and emotional behaviors were assessed, together with long-term depression plasticity in accumbal medium spiny neurons. Overall, mice fed with n-3 PUFA supplementation displayed an emotional behavior profile and electrophysiological properties of medium spiny neurons which was distinct from the ones displayed by mice fed with the control diet, and this, independently of CSDS. Using the social interaction index to discriminate resilient and susceptible mice in the CSDS groups, n-3 supplementation promoted resiliency. Altogether, our results pinpoint that exposure to a diet rich in LC n-3 PUFA, as compared to a diet rich in SC n-3 PUFA, influences the NAc fatty acid profile. In addition, electrophysiological properties and emotional behavior were altered in LC n-3 PUFA mice, independently of CSDS. Our results bring new insights about the effect of LC n-3 PUFA on emotional behavior and synaptic plasticity.

## 1. Introduction

Major depressive disorder (MDD) is a leading cause of disability worldwide, with around 20% of subjects affected in the general population [1]. There are few effective treatments for depression, with current treatments leaving 10–30% of patients with no efficient response [2]. Thus, there is an urgent need to design and develop novel therapeutics to treat depression.

The long-chain (LC) n-3 polyunsaturated fatty acids (PUFAs), eicosapentaenoic acid (EPA, C20:5 n-3) and docosahexaenoic acid (DHA, C22:6 n-3), are considered promising nutritional-based therapeutic strategies for preventing and treating MDD [3,4,5,6]. The use of these fatty acids in MDD is based on the observation of an inverse association between the intake of oily fish (rich in EPA and DHA) and the prevalence of MDDs in humans, as confirmed recently [7,8]. This is also corroborated by the observation that MDD patients exhibit lower blood and brain levels of EPA and/or DHA as compared to healthy controls [9,10]. Importantly, depressive symptoms in patients were reported to be significantly reduced with a dietary supplementation containing more than 50% of EPA, but not with DHA [11]. The potent antidepressant effect of a dietary formulation with a higher amount of EPA to DHA rather than DHA alone in MDD was recently confirmed by a meta-analysis [4], although another study did not observe these effects [12]. The potential antidepressant effect of EPA was also demonstrated in animal models of depression, such as olfactory bulbectomized rats [13], chronic unpredictable mild stress [14], maternal stress [15], or chronic social defeat stress (CSDS) [16]. In particular, we previously demonstrated that a dietary intervention with a diet enriched with LC n-3 PUFA (10% EPA and 7% DHA of total fatty acids, see [17]) partially protected mice from CSDS-induced emotion behavior alteration [16,18]. Altogether, these data suggest that a dietary supplementation with EPA/DHA with a higher proportion of EPA could be efficient to improve symptoms in MDD, however the mechanisms involved are still poorly known.

Some mechanistic explanations of the antidepressant activities of EPA/DHA have been suggested [3,6]. Indeed, it has been shown that LC n-3 PUFA dietary supplementation reduces inflammatory processes, including in the brain [19,20], hypothalamic–pituitary–adrenal (HPA) axis alteration [5,21,22,23,24], apical dendritic tree alterations [16], and promotes neurogenesis [25], which are altered in depression [14]. Although there is a close link between n-3 PUFA, synaptic plasticity, and depressive-like symptoms [6,18,26,27], the effect of EPA/DHA supplementation on synaptic plasticity has been poorly studied. Furthermore, whether EPA/DHA supplementation can restore the synaptic deficit observed in animal models of depression remains to be elucidated. Indeed, we have previously reported that a diet low in n-3 PUFA, which induces a decrease of brain DHA, triggers an endocannabinoid (eCB)-dependent synaptic plasticity impairment in the nucleus accumbens (NAc) and the prefrontal cortex of mice [28,29], together with an altered apical tree of neurons [16] and eCB signaling in the brain [30]. Restoration of the eCB-dependent synaptic plasticity can be achieved through pharmacological approaches aiming at increasing eCBs at the synaptic level, thus improving depressive-like symptoms in n-3 PUFA-deficient mice [28]. These results are in accordance with the observation that accumbal activity is altered in MDD patients [31]. Importantly, eCB-dependent synaptic plasticity is altered in animal models of depression [32,33] and the pharmacological restoration of this plasticity in the NAc alleviates the deleterious effect of CSDS on emotional behavioral [33]. Altogether, these data suggest that eCB-dependent synaptic plasticity in the NAc sustains emotional behavior alterations in animal models of depression, such as CSDS.

Based on previous data, including ours, showing that dietary intervention using diets rich in LC n-3 PUFAs protects from the development of emotional behavior alteration in animal models of chronic stress [16], we analyzed whether a diet supplemented with EPA/DHA controls emotional behavior and eCB-dependent synaptic plasticity in the NAc of mice submitted or not to CSDS. For the first time, we established the PUFA and lipid metabolism molecular profiles in the NAc of mice fed a diet rich in EPA + DHA for 8 weeks, as compared to a control diet balanced in relatively short-chain (SC) n-3 PUFA (α-linolenic acid, ALA), starting at weaning. Then, we examined whether such a diet rich in EPA + DHA could regulate emotional behavior and eCB-dependent plasticity in the NAc of mice submitted or not to CSDS. We identified a lipid metabolic signature in the NAc of EPA + DHA mice which parallels emotional behavior and accumbal electrophysiological properties, providing insight on the effect of diets rich in EPA + DHA on brain function and emotional behavior.

## 2. Results

### 2.1. Fatty Acids and Molecular Signature of LC n-3 PUFA Dietary Supplementation

We first characterized the fatty acid signature in the NAc of mice fed with a relatively short-chain (SC) n-3 PUFA diet (controls) or a LC n-3 PUFA (n-3 suppl) diet after 8 weeks of feeding (Figure 1). 

The LC n-3 PUFA diet induced significant increases in n-3 PUFA content, including DPA (n-3), DHA, and total n-3 PUFAs (Figure 2, Appendix A (see Appendix A), *p* < 0.05, unpaired *t*-tests). In line with these results, significant decreases of n-6 PUFA content were also observed, including linoleic acid (LA C18:2 n-6), arachidonic acid (ARA C20:4 n-6), docosatetraenoic acid (C22:4 n-6), DPA (n-6), and total n-6 PUFAs (Figure 2, *p* < 0.05, unpaired *t*-tests). Interestingly, tetracosatetraenoic acid (C24:4 n-6) was increased following LC n-3 PUFA dietary supplementation (*p* < 0.05, unpaired *t*-tests). As expected, the ratio of n-3/n-6 was also significantly increased (*p* < 0.05, unpaired *t*-test) by the LC n-3 PUFA supplementation.

These results indicate that n-3 PUFAs dietary supplementation can alter fatty acid content in the NAc.

In agreement with the above-mentioned fatty acid modifications observed in the NAc, we observed that gene expression of lipid metabolism pathways was altered by the LC n-3 PUFA dietary exposure, as compared to the control diet. Indeed, in the NAc of mice fed with the LC n-3 PUFA diet, we observed significant increases (*p* < 0.05, unpaired *t*-tests) of gene expression in 25 out of the 89 genes tested in a lipidomic microarray (Figure 3A, Appendix A). These included genes coding for hydrolases (*Abhd4*, *Abhd6*), desydrogenases (*Acadl*, *Acadvl*, *Hadh*), elongases (*Elovl1*, *Elolv5*), synthases (*Ptgs1*, *Fasn*), transferases (*Acat1*, *Pcyt1a*, *Pcyt1b*, *Pcyt2*), as well as fatty acid receptors (*Adipor2*, *Ppard*, *Ppargc1a*, *Ptger4*, *Rxra*) and transporters (*Apoe*, *Fads2*, *Scd1*, *Slc25a20*, *Slc27a1*, *Slc27a3*, *Slc27a4*; Figure 3B). As expected, gene ontology [34] using over-representation analysis, performed with Webgestalt [35,36], identified several biological processes related to fatty acid regulation, such as fatty acid synthesis, transport, and metabolism, as well as lipid biosynthesis/metabolism and cellular lipid metabolism (Figure 3C). Furthermore, due to the design of our microarray and as expected, these gene ontology families presented redundancy (Figure 3D,E). Of note, amongst the top 50 enriched gene ontology families, Ptger4 was associated to several families (8/50), while both *Slc25a20* and Rxra remained unmapped (0/50, not shown).

Altogether, these results suggest that n-3 PUFAs supplementation successfully modulates the NAc fatty acid profile, along with upregulation of genes involved in fatty acid synthesis, metabolism, and transport.

### 2.2. Anxiolytic-like Effect of LC n-3 PUFA Dietary Supplementation

Then, in another cohort of animals, we evaluated the emotional behavior of SC (control) and LC n-3 PUFA mice submitted or not to CSDS, using a battery of tests aimed at assessing stress-related behaviors. We found that CSDS had significant effects (Appendix A) on the time spent by mice in the open field (OF), especially on variables such as time in both the central (F_(1,37)_ = 5.084, *p* = 0.03) and peripheral (F_(1,37)_ = 5.085, *p* = 0.03) zones, as well as total distance travelled (F_(1,37)_ = 6.824, *p* = 0.013, Figure 4A). Similarly, mice submitted to CSDS spent significantly less time in the light zone (F_(1,37)_ = 4.159, *p* = 0.04) and significantly more time in the dark zone (F_(1,37)_ = 4.156, *p* = 0.04) during the light–dark (LD) test (Figure 4B). The number of entries in the light zone or total distance travelled by stressed mice was unchanged. In the elevated plus maze (EPM), CSDS did not induce any difference on all parameters assessed (Figure 4C). Exposure to a LC n-3 PUFA diet 8 weeks before the induction of CSDS induced anxiolytic-like effects in undefeated and defeated mice, as observed in all the variables measured in the OF test (Figure 4A) and the LD test (Figure 4B). In the EPM, LC n-3 PUFA dietary exposure had a significant effect on time spent in the center (F_(1,37)_ = 4.405, *p* = 0.04) and total distance travelled (Figure 4C, F_(1,37)_ = 4.832, *p* = 0.03). No significant interaction was observed between CSDS and LC n-3 PUFA dietary exposure, although it almost reached statistical significance in the number of head dipping in the EPM (*p* = 0.06, Figure 4C). Finally, mice submitted to CSDS presented significantly higher anxiety scores (Figure 4D, F_(1,37)_ = 5.491, *p* = 0.025), as observed in a previous study [33].

These results indicate that n-3 PUFAs dietary supplementation can promote anxiolytic-like effects in mice, irrespective of CSDS.

### 2.3. Reversal of CSDS-Induced Social Interaction Deficits by LC n-3 PUFA Dietary Supplementation

We then measured social interaction in defeated mice fed with either SC (control) or LC n-3 PUFA-supplemented diets. Social interaction ratios were not different across groups (Figure 5A, stress: F_(1,37)_ = 1.337, *p* = 0.25; diet: F_(1,37)_ = 1.007, *p* = 0.32; interaction: F_(1,37)_ = 2.425, *p* = 0.13). In the two groups of mice exposed to CSDS, the proportion of susceptible (ratio < 1) and resilient mice (ratio > 1) [37,38] was different, although it did not reach statistical significance (*p* = 0.33, Fisher’s exact test), likely due to the small sample size (Appendix A, Figure 5B). Resampling at 100 replicates exacerbated the differences observed in both social interaction ratios and the proportion of resilient/susceptible mice (Figure 5C), which was not due to the resampling method itself (Appendix A, Appendix A).

These results suggest that n-3 PUFAs dietary supplementation increases social interaction following CSDS, but not under baseline conditions.

### 2.4. Alteration of Electrophysiological Properties of Accumbal Medium Spiny Neurons by LC n-3 PUFA Dietary Supplementation

We observed a significant effect of the LC n-3 PUFA supplementation on several intrinsic electrophysiological parameters of accumbal medium spiny neurons (MSN). These included significantly altered passive membrane properties, such as an increase in resting membrane potential (RMP, F_(1,37)_ = 5.73, *p* = 0.02) and capacitance (F_(1,37)_ = 9.92, *p* = 0.003), but not input resistance or rheobase (Figure 6A). CSDS and LC n-3 PUFA supplementation induced significant alterations of voltage over current relationships (stress: F_(1,6)_ = 11.68, *p* < 0.001; diet: F_(1,6)_ = 22.37, *p* < 0.001), while only n-3 supplementation had a significant effect on the number of action potentials generated during supra-threshold current applications (Figure 6B, F_(1,16)_ = 41.81, *p* < 0.001).

Furthermore, LC n-3 PUFA-supplemented mice presented a higher action potential (AP) amplitude (F_(1,37)_ = 9.44, *p* = 0.004) and longer AP duration (F_(1,37)_ = 12.00, *p* = 0.001), as well as a shorter delay to first spike (Table 1, Appendix A, F_(1,37)_ = 5.51, *p* = 0.02). These altered spike properties suggest a potential for increased neurotransmitter release by LC n-3 PUFA supplementation.

We then applied an electric protocol (10 Hz stimulation for 10 min) to induce long-term depression (LTD) in accumbal MSN of SC (control) and LC n-3 PUFA-supplemented mice submitted or not to CSDS. In SC (control) (Figure 7A) and LC n-3 PUFA-supplemented (Figure 7B) mice, CSDS did not alter LTD. However, mice fed the LC n-3 PUFA-supplemented diet presented significantly decreased excitatory post-synaptic currents (EPSCs) at baseline (Figure 7C, Appendix A, F_(1,34)_ = 4.361, *p* = 0.04), which were not to be attributed to differences in stimulation intensity (Figure 7D). This was also confirmed by examining input–output relationships (Appendix A, F_(1,53)_ = 4.209, *p* = 0.04). Finally, no differences were observed following the induction of LTD, where early and late EPSC amplitudes were similar across all groups (Figure 7E, Appendix A), suggesting that LC n-3 PUFA supplementation has an effect on excitatory transmission onto accumbal MSN, without altering the expression of LTD in these neurons.

These results suggest that LC n-3 PUFA supplementation can modulate the electrophysiological properties of accumbal MSN, without altering LTD. Finally, the possible correlations between *in vivo* behavioral measurements and *ex vivo* electrophysiological properties of MSN are presented in Appendix A.

## 3. Discussion

First, we identified for the first time the fatty acids and lipid genes signature in the NAc of adult mice fed for 2 months (starting at weaning) with a diet enriched with LC n-3 PUFA (10% EPA and 7% DHA of total lipids) as compared to a SC isocaloric diet rich in ALA (the precursor of EPA and DHA). The specific changes in this signature were accompanied by an alteration of emotional behaviors, as measured in the OF, LD, and EPM tests. Following CSDS, the number of stress-resilient mice fed with control or n-3 LC PUFA-supplemented diets was not significantly different, likely due to the low number of animals used in the test. However, using a bootstrap method aiming at modeling the repartition of stress-resilient and -prone mice suggested that the consumption of a diet enriched in LC n-3 PUFA could promote CSDS resiliency. In addition, the electrophysiological properties of accumbal MSN were altered by diets, independently of CSDS. Overall, our results pinpoint that the consumption of a diet rich in LC n-3 PUFA influences the NAc fatty acid profile, which is likely to have an effect on neurotransmission and emotional behavior.

It is well-known that n-3 PUFA dietary content influences brain fatty acid levels [39]. We and others have previously reported that PUFA levels, in particular DHA and DPA n-6, are decreased and increased, respectively, by a low level of dietary n-3 PUFA in the form of a precursor (ALA) starting at weaning/adolescence, in the NAc and prefrontal cortex (PFC) of rodents [28,40]. Decreasing DHA and increasing DPA n-6 in PFC or NAc are even more pronounced when the low n-3 PUFA diet starts at developmental stages [29,41]. In this work, we describe for the first time that a two-month-long dietary intervention starting at weaning with diets either rich in SC (ALA) or rich in LC (EPA + DHA) n-3 PUFAs had a significant effect on NAc fatty acid profiles, with a decrease of several n-6 PUFA species and an increase of several n-3 PUFA species, namely DPA n-3 and DHA, leading to strong changes in the n-3/n-6 PUFA ratio in this structure. This suggests that a short-term dietary intervention has an impact on the NAc fatty acid signature, which corroborates previous results obtained in other brain structures, such as the hippocampus [40,42]. In addition, a molecular approach allowing to measure the expression of almost 90 genes involved in fatty acid metabolism, activity, and transport revealed a specific molecular signature in the NAc of LC n-3 PUFA-supplemented mice. Among the genes for which expression was increased by the dietary exposure to LC n-3 PUFA, some are directly linked to n-3 PUFA transport, metabolism, and activity. Indeed, elongation of very-long-chain fatty acid (ELOVL) enzymes are involved in both long and very LC fatty acid metabolism, including EPA and DHA [43], and have been reported to be expressed in neurons and glial cells [44,45]. FADS2 (Δ6 fatty acid desaturase, the rate-limiting enzyme of the LC PUFA biosynthesis) mRNA expression is also increased in the NAc of LC n-3 PUFA mice. As FADS2 is expressed in astrocytes, and not neurons, it has been suggested that, in addition to the plasma DHA pool, these cells could supply neurons with DHA [46,47]. Interestingly, in humans, some FADS genetic variation has been associated to a lower DHA bioavailability and to MDD [48]. Furthermore, FADS1 expression was shown to be reduced in the brain of patients with depression [49]. In vitro, DHA triggers a decreased expression of FADS2 in neurons and astrocytes [50]. However, in the presence of retinoic acid acting through PPAR and RXR receptors, we found that their expression increased in the NAc of LC n-3 PUFA (EPA + DHA) mice as DHA activates FADS2 expression [47]. The solute carrier 27A (SLC27A) gene family encodes fatty acid transport proteins (FATPs), some of which were found here to be upregulated in the NAc of LC n-3 PUFA-supplemented mice. SLC27A3 has been reported to be highly expressed in several organs, including the neonatal and adult brain [51]. In addition, SLC25A20, also known as carnitine acyl-carnitine carrier (CAC), is a mitochondrial carrier involved in fatty acid β-oxidation [52] which was highly expressed in the NAc of mice fed with LC n-3 PUFA (EPA + DHA). Fish oil supplementation has been reported to decrease CAC expression in the liver [53], but tissue expression of CAC varies in different tissue, which has been reported as very low in the brain [54]. Altogether, the increased expressions of genes belonging to LC PUFAs biosynthesis, transport, and activity are in adequation to the effects of increased LC n-3 PUFA following dietary intake.

The mRNA expression of *Ptger4*, one of the receptors of prostaglandin E2, is increased in the NAc of LC n-3 PUFA-supplemented mice. This receptor is expressed in neurons and microglia and has been previously shown to be neuroprotective in the context of brain inflammation, neurodegeneration, and lesions [55,56,57,58]. A compensatory mechanism between decreased ARA (n-6 PUFA, precursor of prostaglandin E2 [19]) and increased expression of *Pterg4* could be suspected in our experiment. Interestingly, in a non-inflammatory context, the lack of EP4 signaling is associated with increased depression-like behavior [59], suggesting that this receptor could play a role in emotional behavior, as it has been described for other prostaglandin receptors (EP2, EP3) which also mediate the prostaglandin effect on synaptic plasticity in the spinal cord [60]. However, whether *Ptger4* is involved in the effect of LC n-3 PUFA (EPA + DHA) on emotional behavior remains to be determined.

CSDS is known to induce social and emotional behavior alteration that can segregate resilient *vs*. susceptible mice [37,38,61]. Using a segregation index based on social interaction, we did not observe any significant difference in the number of susceptible and resilient mice according to the diets, which can be attributed to the low number of animals used in the present study (n = 7–10) compared to previous studies, in which between 70 to 437 animals were used [37,38,62,63,64]. We therefore undertook a bootstrapping strategy [65] aiming at modeling the repartition of stress-resilient and -prone mice based on the results obtained in 7–10 animals/group used in this study in accordance with local and national ethical procedures, in particular the 3R rule. Interestingly, the bootstrapping strategy suggested that LC n-3 PUFA dietary consumption could promote CSDS resiliency, as illustrated by the spread of the confidence intervals following 100 bootstrap runs (see Figure 5). Using such a strategy, we observed that LC n-3 PUFA yielded a 95% CI of the social interaction ratio strictly above 1 (1.16–1.22), indicative of stress resiliency [37,38,61] in the majority of animals. Concerning emotional behaviors, as measured by OF, LD, and EPM, we found that CSDS significantly alters several parameters which are signs of anxiety (time in the central zone (OF) or in the light zone (LD)). This suggests that anxiety-like behaviors are altered by chronic stress, as previously reported in mice fed with a regular diet [33]. In contrast, the LC n-3 PUFA dietary intervention increases the time spent in the anxious zone, as measured in OF and LD tests, suggesting that the consumption of LC n-3 PUFA promotes non-anxious behavior independently of chronic stress. However, no interactions between stress and diet were revealed. Whether the changes in PUFA levels and/or lipid metabolism in the NAc are involved in the anxiolytic effect of the consumption of a diet rich in EPA + DHA remains to be determined.

Altered emotional phenotypes are paralleled to significant electrophysiological impairments in accumbal MSN, as observed in these neurons following resilience or susceptibility to stress [33,61], likely arising from neuronal morphology remodeling [16,66]. CSDS-induced electrophysiological alterations in the NAc have been observed before [33]. However, these results were acquired on animals fed with a standard laboratory diet (A04), a diet in which the ratio of LA to ALA was 15:1, which can explain the differences observed between the current study and the previous study. In contrast to LC n-3 PUFA supplementation, our laboratory has also investigated the electrophysiological consequences of n-3 deficiency. Following dietary n-3 PUFA dietary deficiency, n-3 PUFA-deficient mice presented abnormal endocannabinoid-dependent plasticity in the NAc, which could be restored via pharmacological enhancement of mGluR_5_ or by increasing the levels of the endocannabinoid 2-arachidonoylglycerol [28]. Our previous studies have also witnessed the presence of endocannabinoid-dependent plasticity in the brain of mice fed with diets balanced in n-3 PUFAs. This was observed in different brain regions, such as the prefrontal cortex [28,29] and the hippocampus [67], which are also both impaired following dietary n-3 PUFA deficiency. The present study also identified such a plasticity in the NAc of mice fed with a balanced diet. However, we observed a decreased neuronal response following LC n-3 PUFA dietary supplementation, together with altered intrinsic properties, suggesting that n-3 PUFA content in the brain finely tunes neuronal integration [27]. Most electrophysiological properties were affected by LC n-3 PUFA dietary supplementation, while only a few were affected by chronic stress. These observations are in line with the altered emotional behavior of animals following LC n-3 PUFA dietary intake, while stress did not produce such drastic effects on emotional behavior. In the present study, we have demonstrated that dietary intake of LC n-3 PUFAs can modulate emotional behaviors and synaptic plasticity in the NAc. Whether these effects are in a direct relationship remains to be elucidated.

In the present study, only male mice were used. Indeed, we used male CD1 Swiss retired breeders as aggressors, due to the aggressive behaviors displayed by these animals. To avoid mating, female C57Bl/6j mice were not used in our paradigm. While we acknowledge that CSDS can be performed in females, using different experimental protocols [68,69,70,71], this was outside of the scope of our study. Another potential limitation resides in the fact that we did not assess the relationship between accumbal long-term potentiation (LTP), CSDS, and n-3 PUFAs dietary supplementation. On the one hand, it was previously reported that DHA can modulate hippocampal LTD [72] and that maternal n-3 PUFAs dietary intake could promote LTP in the hippocampus of rat pups [73]. On the other hand, previous studies have demonstrated that CSDS can impair LTP in the hippocampus of mice [74] and rats [75,76,77]. To our knowledge, there are no studies that assessed accumbal LTP in the context of n-3 PUFAs dietary supplementation and CSDS, which will be interesting topics to address in future studies.

## 4. Material and Methods

### 4.1. Ethical Approval

All experiments were performed in accordance with local Ethics policies. The study was approved by the French Ministry of Education and Research (Ministère de l’Education Nationale, de l’Enseignement Supérieur et de la Recherche, agreement number 2018102215303008-V5-APAFiS-17200) following initial validation by the Ethics Committee of the University of Bordeaux (CEEA50).

### 4.2. Animals

Animals were housed under standard housing conditions in a temperature- (23 ± 1 °C) and humidity (40%)-controlled animal room facility, with a 12 h light/dark cycle (07:00–19:00 h), in polysulfone cages (42.5 × 26.6 × 18.5 cm) with *ad libitum* access to water and experimental diet. Fifty-four C57Bl/6j (three weeks old) and fourteen (retired breeder) Swiss CD1 mice were purchased from Janvier Labs (France). All mice were male. Retired breeder Swiss CD1 mice were used as aggressors in the CSDS paradigm and were thus housed individually, while C57Bl/6j mice were housed in pairs.

### 4.3. Diets

Composition of the SC n-3- (“control”) and LC n-3-supplemented (“n-3 suppl”) diets were described previously [16,17] and were manufactured at the INRAE unit at Jouy-en-Josas, France. Diets were started at weaning (P21, Figure 1A,B), as these previous studies have shown that a two-month supplementation with a diet enriched in tuna oil increased DHA levels in the brain [16,17]. Dietary n-3 PUFA supplementation (n-3 suppl) consisted of an isocaloric diet containing 6% of tuna oil (rich in EPA (20:5 n-3) and DHA (22:6 n-3)), while the control diet (control) consisted in 6% of rapeseed oil (rich in ALA (18:3n-3)). Proportions of EPA and DHA in the n-3-supplemented diet were 10% and 7%, respectively [17]. These diets contained similar quantities of casein, starch, cellulose, sucrose, lipids, and minerals [16,17]. Diets were maintained throughout, from P21 until the end of the experiments (biochemistry or electrophysiology, Figure 1A,B, respectively).

### 4.4. Chronic Social Defeat Stress (CSDS)

CSDS was performed as previously described [16,33]. Briefly, SC (control) and LC (n-3 suppl) mice were placed in contact with an aggressor (Swiss CD1 mouse) for 5 min, followed by 4 h of sensorial interactions (no physical contact), every day and for 10 consecutive days (Figure 1C).

### 4.5. Behavioral Assessments

Following the last day of CSDS, mice were scored on four different anxiety-related tests, which were performed over 48 h, as detailed previously [16,29], and during the light phase. These included an OF test (10 min, 300 lux), a social interaction test (no-target/target, 5 min each, 30 lux), a LD box test (8 min, 300 lux), and an EPM test (5 min, 15 lux). Each apparatus was cleaned between experimental runs to avoid olfactory disturbances. Run orders were randomized. Video tracking was used for *a posteriori* analyses.

Anxiety scores were calculated as previously explained [33]. Mice were assigned individual anxiety scores, calculated as the algebraic sum of normalized scores for each of the 6 analyzed anxiety-related behavior tests: (i) time spent in the center (s) of the open field, (ii) time spent (s) and (iii) number of entries in the light compartments of the light/dark box, (iv) time spent (s) and (v) number of entries in the open arms of the elevated plus maze, and (vi) number of head dipping in the risk zone of the open arms in the elevated plus maze. When more than one parameter was used for a behavioral test, we weighted each component accordingly so that each behavioral test weighted 100%. This procedure yielded scores distributed along a 0–3 scale, with 3 reflecting high anxiety. For details about normalization, please refer to Section 4.10 below (data analysis).

### 4.6. Gene Expression Analysis

Accumbal RNA from SC (control) and LC n-3-supplemented mice were extracted using TRIzol extraction kit (Invitrogen, Life Technologies, Saint-Quentin-Fallavier, France). Purity and concentration of RNA were determined using a Nanodrop 1000 spectrophotometer (Nanodrop technologies, Wilmington, DE, USA) and a bioanalyzer (Agilent, Les Ulis, France). Gene expression profiles were performed at the GeT-TRiX facility (GenoToul, Toulouse, France) using Agilent Sureprint G3 Mouse microarrays (8 × 60 K, design 074809), as described earlier [78]. For each sample, Cyanine-3 (Cy3)-labeled cRNA was prepared from 25 ng of total RNA using the One-Color Quick Amp Labeling kit (Agilent), followed by Agencourt RNAClean XP (Agencourt Bioscience Corporation, Beverly, MA, USA). Dye incorporation and cRNA yield were checked using the Dropsense 96 UV/VIS droplet reader (Trinean, Gent, Belgium). Then, 600 ng of Cy3-labeled cRNA was hybridized on the microarray slides. Immediately after washing, slides were scanned on an Agilent G2505C Microarray Scanner using Agilent Scan Control A.8.5.1 software, and the fluorescence signal was extracted using Agilent Feature Extraction software v10.10.1.1 with default parameters. Raw data (median signal intensity) were filtered, log2-transformed, corrected for batch effects (microarray washing bath and labeling serials), and normalized using the quantile method. All genes analyzed in the present study are listed in Appendix A, together with respective forward and reverse primer sequences in Appendix A.

### 4.7. Endogenous Fatty Acid Determination by Gas Chromatography-Flame Ion Detection

Tissue fatty acid content was determined as previously described [79]. Briefly, total lipid extracts were obtained by following the method of Folch, Lees, and Stanley [80], consisting in lipid extraction in a mixture of chloroform:methanol:potassium chloride 0.88% (2:1:0.8 by volume). After vortexing and centrifugation at 500× *g* (10 min), the chloroform-containing layer was pipetted into a new tube. Helium-driven gas chromatography (Varian 430, Bruker, San Jose, CA, USA) was used to determine the samples’ (1 µL) fatty acid tissue content by comparing retention times on a DB-23 (50% cyanopropyl)-methylpolysiloxane capillary column (J&W Scientific, Agilent Technologies, Mississauga, ON, Canada) with external standard samples (GLC-674, Nu Chek Prep Inc., Elysian, MN, USA). Detection was performed using a flame ionization detector (300 °C) with air (300 mL/min) and helium (29 mL/min), with sampling rates at 20 Hz.

### 4.8. Ex Vivo Whole-Cell Patch-Clamp Electrophysiology

Animals were deeply anaesthetized using isoflurane inhalations, followed by quick decapitation. Striatal slices, containing the NAc, were cut using a VT1000S vibratome (Leica, Germany) at a thickness of 350 μm in ice-cold artificial cerebrospinal fluid (ACSF) containing (in mM): 125 NaCl, 2.5 KCl, 1.25 NaH_2_PO_4_, 2.0 CaCl_2_, 1.0 MgCl_2_, 25 NaHCO_3_, and 25 glucose (osmolarity 308 ± 3 mOsm), bubbled continuously with carbogen (95% O_2_ and 5% CO_2_). Slices (4–6 per animal) were then transferred to a holding chamber, incubated at 32–34 °C for 60 min, and returned to room temperature. For the slicing procedure and the resting of the slices, 10 µM of pyruvic acid was added to the extracellular solution. During recording, slices where continuously perfused at 1.5–2.0 mL/min with oxygenated ACSF. Whole-cell patch-clamp recordings were performed at room temperature (20–22 °C) with electrodes (1.5 mm outer diameter) fabricated from filamented thick-wall fire-polished borosilicate-glass (Sutter Instruments, Novato, CA, USA), pulled using a gravity puller (Narishige, Tokyo, Japan). Pipette resistance was typically between 4 and 7 MΩ when filled with a potassium gluconate-based intracellular solution, consisting of (in mM): K-gluconate 128, NaCl 20, MgCl_2_ 1, EGTA 1, CaCl_2_ 0.3, Na_2_-ATP 2, Na-GTP 0.3, cAMP 0.2, and HEPES 10, pH = 7.35, and osmolarity 296 ± 3.8 mOsm. The junction potential was +15.4 mV. Pipette offset was zeroed before each recording.

MSN of the NAc were visualized under direct interference contrast with a BX51WI microscope (Olympus, Tokyo, Japan), mounted on an air table (TMC, Saratoga Springs, NY, USA) and under a Faraday cage, with an upright microscope, a 40× water immersion objective combined with an infra-red filter, a monochrome CCD camera (Roper Scientific, Vianen, The Netherlands), and a PC-compatible system for analysis of images as well as contrast enhancement. Recording pipettes were slowly advanced towards individual striatal MSN within the slice under positive pressure and visual control. On contact, tight gigaohm (GΩ) seals were achieved by applying negative pressure. The membrane patch was then ruptured by suction and both membrane current and potential were monitored using Multiclamp700B and Digidata 1440A by Axon Instrument (Molecular Devices, San Jose, CA, USA). Whole-cell access resistances, measured in voltage clamp, were in the range of 8–20 MΩ. A concentric bipolar electrode (Phymep, Paris, France) was placed on afferent fibers to evoke EPSCs (at 0.1 Hz), recorded in MSN under voltage-clamp configuration with membrane potential clamped at −70 mV. Current over voltage (I/V) curves were acquired in I = 0 (free) current-clamp mode. All data were sampled at 20 kHz and filtered at 1 kHz. Series resistance was measured throughout the experiment with a −5 mV step lasting 50 ms. Synaptic events were stored by using p-CLAMP 10.6 (Axon Instruments, Burlingame, CA, USA) and *a posteriori* analyzed offline on a computer with Axograph software (Axon Instruments, Burlingame, CA, USA). Only data from putative GABAergic MSN were included in the present study, identified immediately after rupture of the GΩ seal, by evaluating responses to the injection of both hyperpolarizing and depolarizing currents, as described previously [81].

LTD was elicited by stimulating afferent fibers at 10 Hz for 10 min, as described before [82,83,84,85].

### 4.9. Resampling

Resampling (bootstrapping) was performed in R [86] with the *boot* package, available at CRAN (Comprehensive R Archive Network; https://cran.r-project.org, accessed on 1 May 2022). Resampling methods, applied to biology, have been explained recently [65]. The visual inference tool (VIT), a component of iNZight [87,88], was used to track the distribution of means from resampling replicates.

### 4.10. Data Analyses

All statistical tests were performed with R [86]. Significance was set at alpha = 0.05. All values are reported as mean ± standard error of the mean (SEM). Data analysis was performed using one-sample *t*-tests, two- or three-way ANOVA, followed by *post hoc* tests if appropriate. Appendix A presents all statistical analysis results performed in the present study. Normalization of behavioral parameters was achieved by using the following formula: $x_{n}=\frac{x-min}{\max-min}$, where $x_{n}$ is the normalized variable value, *max* is the variable maximum value, and *min* is the variable minimum value. Gene expression was calculated relative to the housekeeping gene (Gapdh) and normalized to control levels. Correlation matrices and heatmaps were calculated and drawn in R with the following packages: *colourvalues*, *corrplot*, *ggcorrplot*, *ggplot2*, *gplots*, *heatmap*.*plus*, *heatmap3*, *Hmisc,* and *RColorBrewer*. These packages are all accessible in CRAN (see above, Section 4.9). Enrichment ratio (ER) was calculated by the formula: $ER=\frac{\left(\frac{m}{M}\right)}{\left(\frac{n}{N}\right)}=\frac{\left(m\;x\;N\right)}{\left(M\;x\;n\right)}$, where *m* is the number of mapped genes in the pathway, *N* is the total number of genes within the *mus musculus* genome (~18,985), *M* is the total number of genes in the pathway, and *n* is the number of genes found significantly upregulated in our dataset (= 25). For gene ontology (GO), Webgestalt (http://www.webgestalt.org/, accessed on 1 May 2022) was used [35]. All figures were optimized for color blindness. Box and whiskers and violin plots were plotted from minimum to maximum values. Please refer to Appendix A for body weights and food consumption. Food consumption was estimated by averaging food intake over the duration of the experimental protocol, divided by the number of animals per cage.

## Figures and Tables

**Figure 1 ijms-23-06650-f001:**
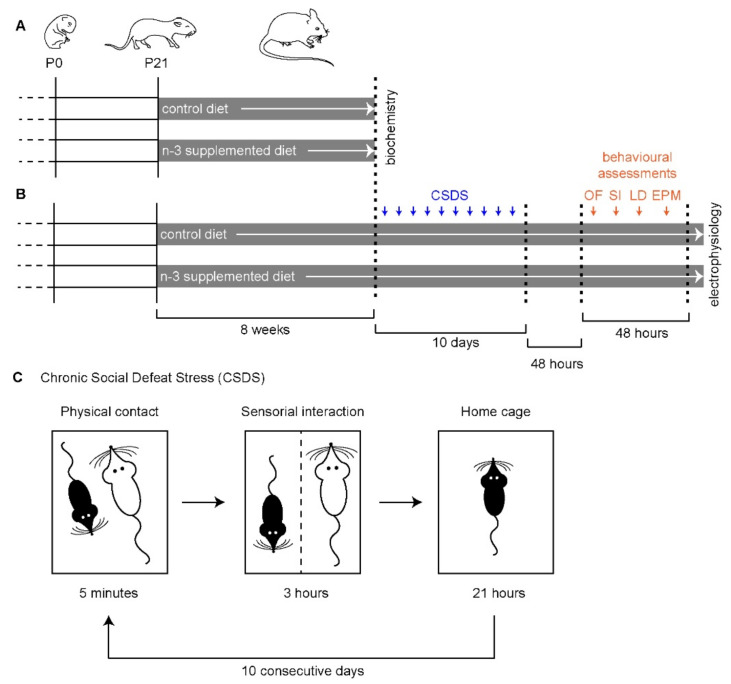
Experimental protocol used in the present study. (**A**) Control and n-3-supplemented diets were given for 8 weeks to C57Bl6/j mice, starting at weaning (P21), before performing biochemical analyses (striatal fatty acid content). (**B**) Following 8 weeks of diet, chronic social defeat stress (CSDS) was performed every day for a total of 10 consecutive days, which was followed by 48 h of behavioral testing, including the open field (OF), the social interaction (SI) test, the light–dark (LD) test, and the elevated plus maze (EPM) test. Two tests were performed per day, in the light phase. Following 48 h of rest, mice were tested for endocannabinoid-dependent plasticity (electrophysiology). (**C**) Detail of the CSDS procedure. Defeated mice are represented in black, while the Swiss CD1 aggressor is represented in white. Drawings are not to scale.

**Figure 2 ijms-23-06650-f002:**
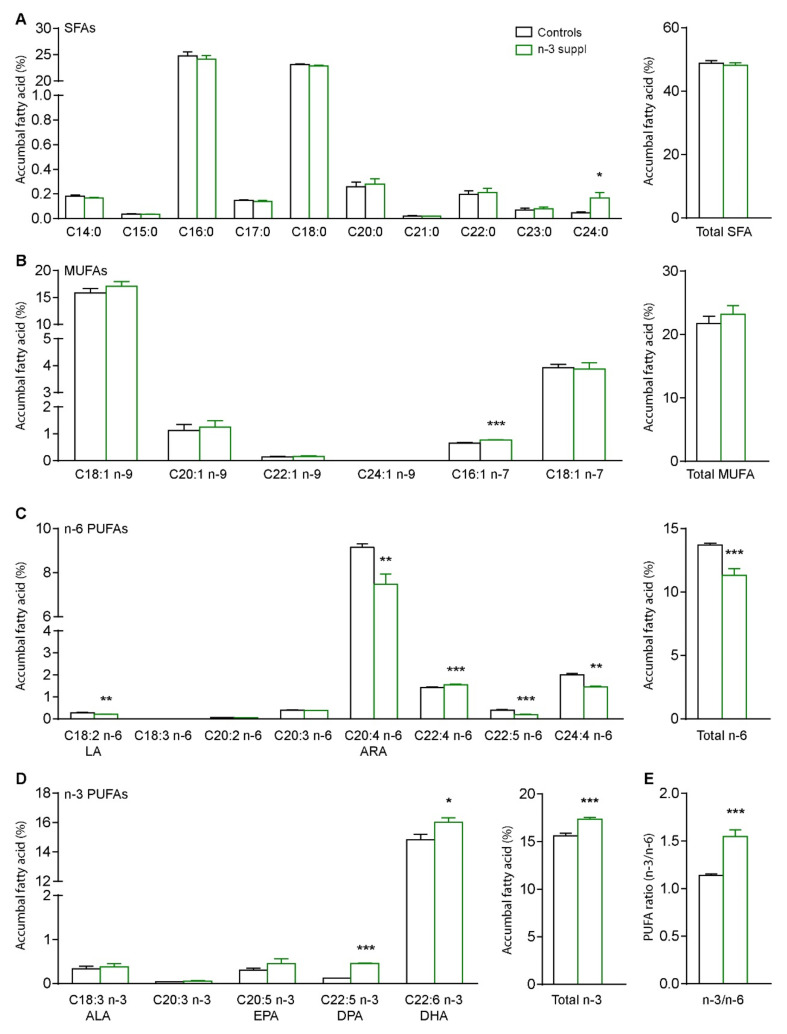
Accumbal fatty acid profiles of control and n-3-supplemented (n-3 suppl) mice. (**A**) Saturated fatty acids (SFA). (**B**) Mono-unsaturated fatty acids (MUFAs). Polyunsaturated fatty acids (PUFAs) of the n-6 (**C**) and n-3 (**D**) families. (**E**) Ratio of n-3 over n-6 PUFAs. LA: linoleic acid, ALA: alpha-linolenic acid, ARA: arachidonic acid, EPA: eicosapentaeonic acid, DPA: docosapentaenoic acid, DHA: docosahexaenoic acid. Unpaired *t*-tests. * *p* < 0.05, ** *p* < 0.01, and *** *p* < 0.001 *vs*. control. Histograms represent mean ± SEM.

**Figure 3 ijms-23-06650-f003:**
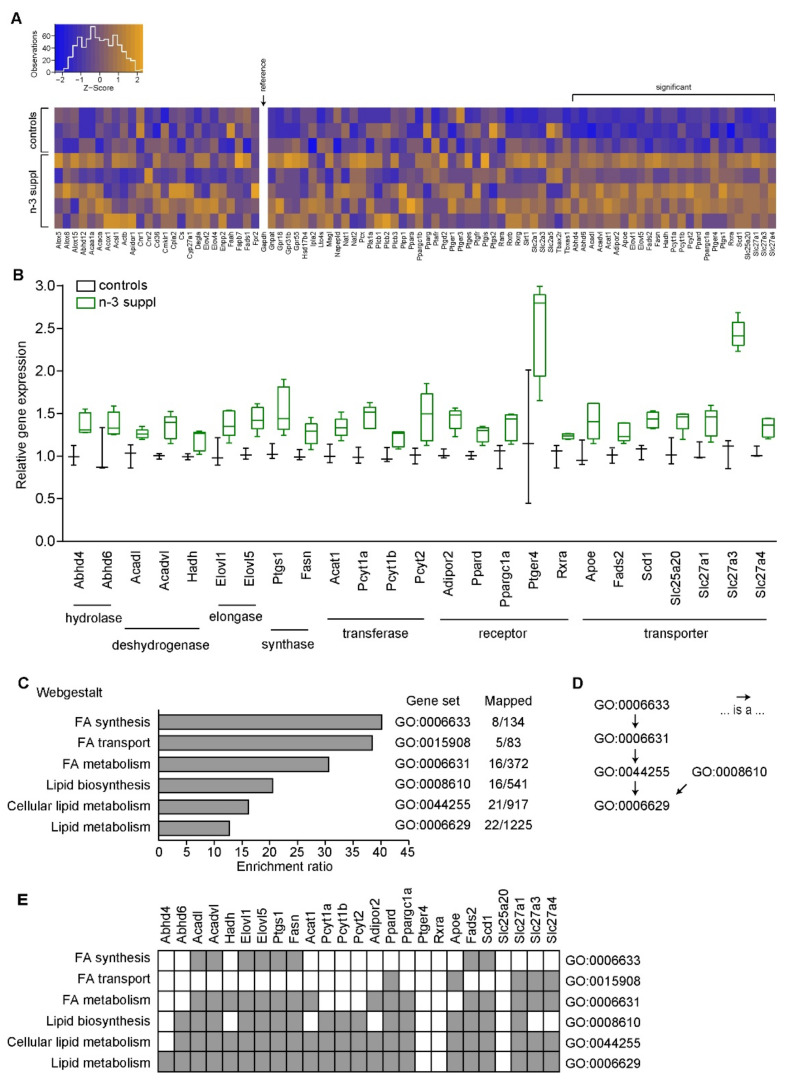
n-3 supplementation increases transcription of genes involved in fatty acid processes. (**A**) Heatmap of transcriptional regulation of the 89 analyzed genes presenting significant or non-significant gene modifications. Gapdh is used as the housekeeping gene (reference). Insert represents Z-scores’ distribution. (**B**) Significantly upregulated genes. Values (box and whiskers plots) are plotted from minimum to maximum. (**C**) Gene ontology (Webgestalt [35,36]) and enrichment ratios of the significantly upregulated genes induced by n-3 supplementation. Such an analysis yielded different gene ontology (GO) families, represented with their respective GO codes. Mapped terms within our study are indicated over the total number of genes within each GO family. (**D**) Redundancy of the gene ontology families. Arrows indicate inclusive relationships. (**E**) Detailed mapping of genes and ontology families. Genes were found to be either present (grey) or absent (white) from different gene ontology families. Ptger4 was linked to 8 families from the top 50 enriched ones, while both *Rxra* and *Slc25a20* were not mapped amongst the top 50 enriched terms.

**Figure 4 ijms-23-06650-f004:**
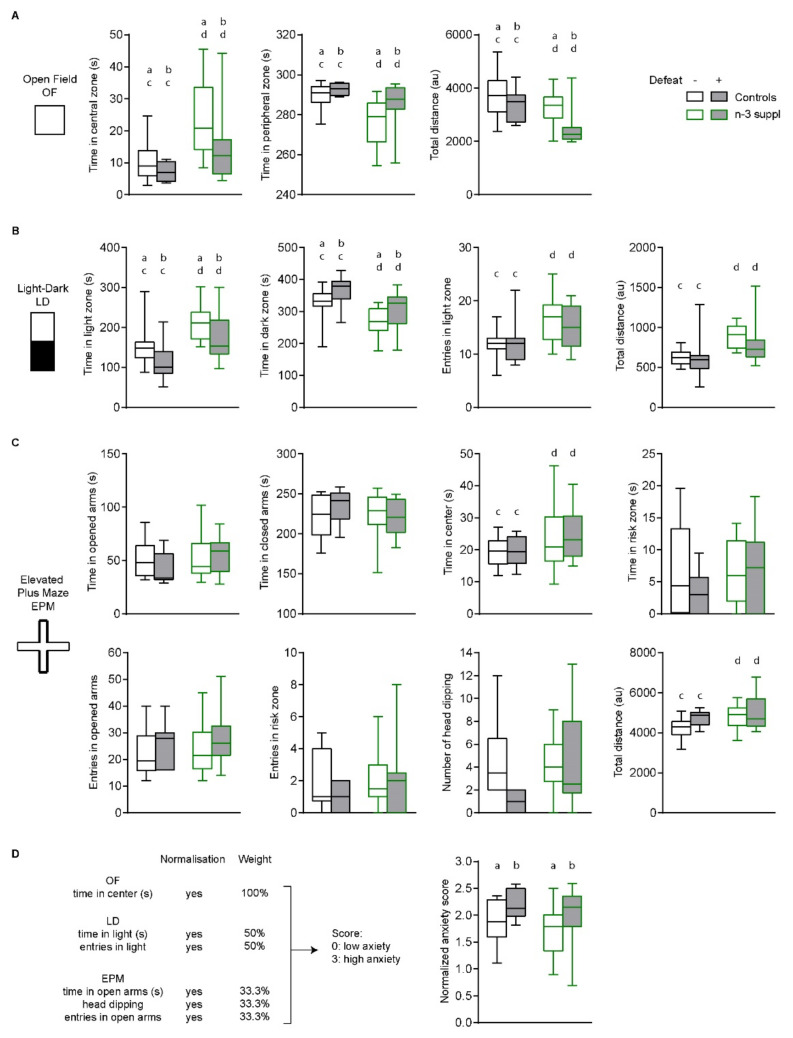
n-3 supplementation has anxiolytic effects. n-3 supplementation induces anxiolytic-like effects in the open field (OF) (**A**) and the light–dark (LD) tests (**B**) but has only minimal effects on the elevated plus maze (EPM) test (**C**). (**D**) Anxiety scores were calculated by normalizing and balancing 6 different behavior measurements. Two-way ANOVAs were performed, with ‘stress’ and ‘diet’ as factors. Statistically significant results are reported as follows: [stress] a *vs.* b, [diet] c *vs.* d. Please refer to Appendix A for further details. Values (box and whiskers plots) are plotted from minimum to maximum.

**Figure 5 ijms-23-06650-f005:**
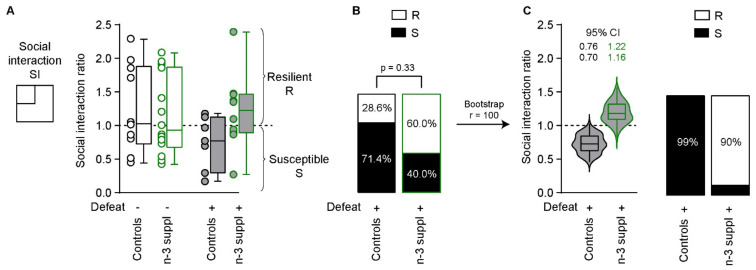
Resilience to CSDS by n-3 supplementation. (**A**) Stress susceptibility (S) or resilience (R) following CSDS or not in control and n-3-supplemented mice. (**B**) Proportions of S or R mice in the two groups. (**C**) Bootstrapping with 100 replicates indicates significant anxiolytic-like effects of the n-3 supplementation on social interaction scores and proportions of S *vs*. R mice in the different groups. Bootstrapping *per se* does not induce statistical significance when replicates are set at 100 (r = 100) (Appendix A). CI: confidence interval. The dotted horizontal bars in (**A**) and (**C**) indicate thresholds for susceptibility (<1) or resilience (>1) to chronic stress, according to Golden and colleagues [37]. The box and whiskers plot (**A**) and the violin plot (**C**) represent values from minimum to maximum. The box and whisker plot within the violin plot is set to represent the 2.5th to 97.5th percentiles.

**Figure 6 ijms-23-06650-f006:**
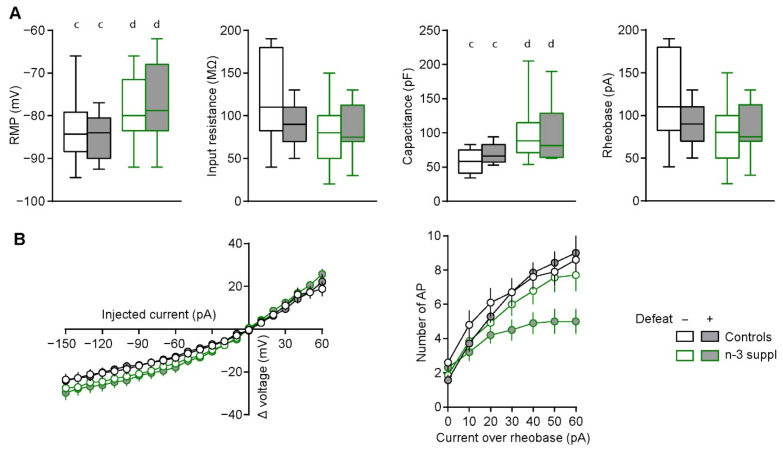
Altered intrinsic electrophysiological properties in accumbal medium spiny neurons induced by n-3 supplementation. (**A**) Resting membrane potential (RMP) and capacitance, but not input resistance or rheobase, were significantly increased following dietary n-3 supplementation. (**B**) Voltage over current (I/V) and number of action potential (AP) generated over rheobase current (I–F) relationships. Stress and n-3 supplementation induced significant effects (Appendix A). Two- and three-way ANOVAs were performed. Statistically significant results are reported as follows: [diet] c *vs*. d. Further statistical details are included in Appendix A. Values (box and whiskers plots) are plotted from minimum to maximum (**A**) or as the mean ± SEM (**B**).

**Figure 7 ijms-23-06650-f007:**
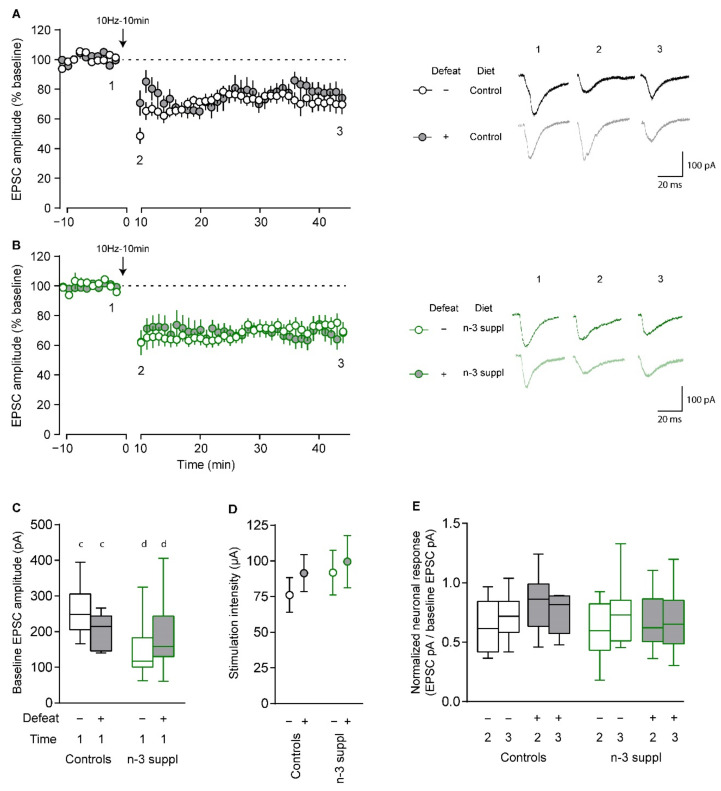
Reduced responses to electrical stimulation in accumbal medium spiny neurons. Representative recording examples of endocannabinoid-dependent plasticity induced by 10 Hz stimulations of pre-synaptic fibers for 10 min in either control (**A**) or n-3-supplemented (**B**) animals. (**C**) Excitatory post-synaptic currents (EPSC) recorded at baseline of recording were significantly lower in n-3-supplemented animals. (**D**) Applied stimulation intensities did not vary across groups. (**C**,**D**) Two-way ANOVAs were performed, with ‘stress’ and ‘diet’ as factors. (**E**) Normalized neuronal responses after LTD (10 Hz, 10 min protocol). Here, normalization was achieved for each neuron by dividing EPSC responses (pA) by baseline EPSC responses (pA). Statistically significant results are reported as for diet, using c *vs*. d. Please refer to Appendix A for further details. Values are plotted as the mean ± SEM (**A**,**B**,**D**) or as minimum to maximum (**C**,**E**).

**Table 1 ijms-23-06650-t001:** Electrophysiological spike properties in medium spiny neurons of control and n-3-supplemented (n-3 suppl) mice. Values are rounded to 1 decimal point and are represented as the mean ± SEM. AP: action potential, CSDS: chronic social defeat stress, SEM: standard error of the mean. Refer to Appendix A for additional details of the two-way ANOVA.

Electrophysiological Parameter	Control	Control + CSDS	n-3 Suppl	n-3 Suppl + CSDS	Significant Variable
AP threshold (mV)	−36.8 ± 1.9	−37.6 ± 2.7	−40.7 ± 0.9	−39.2 ± 1.9	-
AP amplitude (mV)	61.0 ± 2.9	62.3 ± 3.1	72.0 ± 2.1	69.9 ± 3.8	diet
AP duration (ms)	5.6 ± 0.5	5.1 ± 0.4	6.5 ± 0.5	7.8 ± 0.5	diet
Delay to first spike (ms)	387.5 ± 44.2	427.8 ± 55.7	352.4 ± 40.4	253.2 ± 34.8	diet
AP rise kinetics (mV/ms)	34.8 ± 3.3	35.0 ± 3.9	31.5 ± 2.9	27.0 ± 4.3	-
AP decay kinetics (mV/ms)	18.3 ± 2.2	21.2 ± 3.0	19.4 ± 1.3	15.5 ± 1.5	-

## Data Availability

The data presented in this study are available upon request from the corresponding author.

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
