# Peer review of "Dietary Long-Chain n-3 Polyunsaturated Fatty Acid Supplementation Alters Electrophysiological Properties in the Nucleus Accumbens and Emotional Behavior in Naïve and Chronically Stressed Mice"

_ijms, 2022, doi:10.3390/ijms23126650_

Round 1
Reviewer 1 Report
Dietary long-chain n-3 polyunsaturated fatty acid supplementation alters electrophysiological properties in the nucleus accumbens and emotional behavior in naïve and chronically stressed mice.
By Di Miceli et al.
Di Miceli and colleagues investigated the effect of consuming a diet rich in eicosapentaenoic acid (EPA) and docosahexanoic acid (DHA) on emotional behaviors and neuronal activity in the nucleus accumbens (NAc) of mice exposed to chronic social defeat stress. The authors found that anxiety-related behaviors in the open filed test and the light-dark test were decreased in mice fed the EPA- and DHA-rich diet compared to mice fed an alpha-linolenic acid (ALA)-rich isocaloric diet, although any significant impacts were not detected on behavioral outcomes in these tests by exposure to chronic social defeat stress. Combining with the bootstrapping strategy, they also found that stress-resilient mice in the social interaction test were increased by exposure to the EPA- and DHA-rich diet compared to exposure to the ALA-rich diet. The authors further found that several electrophysiological properties of the medium spiny neurons (MSNs) in the NAc were altered by exposure to the EPA- and DHA-rich diet compared to exposure to the ALA-rich diet.Overall, the experiments are well-executed and the findings are important to the field. One potential weakness of this study could be that relationship between increased anxiety-related behaviors and altered electrophysiological properties of the MSNs in the NAc was not investigated; however, the authors are well aware of this potential pitfall and addressed it clearly in the Discussion. Some concerns should be addressed properly in order to be ready for publication.
Comments
1) The authors need to investigate body weight of mice and the amounts of diet consumed. These data are important to estimate how much fatty acids the mice consumed.
2) The authors found increased anxiety-related behaviors in the open field and light-dark tests but not in the elevated plus maze test. It would be better to discuss why such inconsistent behavioral consequences were observed between the behavioral tests. Especially, I am wondering the reason they performed the open field and light-dark tests under relatively bright environment (at 300 lux) but the elevated plus maze test under dim environment (at 15 lux). Could such difference in the floor brightness be account for aforementioned behavioral inconsistency?
3) Did the exposure to the EPA- and DHA-rich diet increase the social interaction ratio in naive mice? This would be useful to consider whether the exposure to the EPA- and DHA-rich diet increases the stress-resilience mice or enhances sociability of mice.
4) The authors need to justify that only males were used in this study or discuss this limitation.
Author Response
Reviewer 1
Di Miceli and colleagues investigated the effect of consuming a diet rich in eicosapentaenoic acid (EPA) and docosahexanoic acid (DHA) on emotional behaviors and neuronal activity in the nucleus accumbens (NAc) of mice exposed to chronic social defeat stress. The authors found that anxiety-related behaviors in the open filed test and the light-dark test were decreased in mice fed the EPA- and DHA-rich diet compared to mice fed an alpha-linolenic acid (ALA)-rich isocaloric diet, although any significant impacts were not detected on behavioral outcomes in these tests by exposure to chronic social defeat stress.
Combining with the bootstrapping strategy, they also found that stress-resilient mice in the social interaction test were increased by exposure to the EPA- and DHA-rich diet compared to exposure to the ALA-rich diet. The authors further found that several electrophysiological properties of the medium spiny neurons (MSNs) in the NAc were altered by exposure to the EPA- and DHA-rich diet compared to exposure to the ALA-rich diet. Overall, the experiments are well-executed and the findings are important to the field. One potential weakness of this study could be that relationship between increased anxiety-related behaviors and altered electrophysiological properties of the MSNs in the NAc was not investigated; however, the authors are well aware of this potential pitfall and addressed it clearly in the Discussion.
We would like to thank both reviewers for their insightful and positive reviews of our manuscript. We highly appreciate the inputs which, we believe, significantly improved the quality of our manuscript. You will find below our responses to the questions raised. The changes have been highlighted in yellow in the updated version of the manuscript.
We would like to thank this reviewer for their very positive reviewing of our work and for raising that the results are of interest to the field.
Concerning the relationship between increased anxiety-like behavior and electrophysiological properties of MSN, we conducted a new analysis using a correlation matrix between all results acquired in the present study, as animals undergoing behavioral assessments were also used for electrophysiological analyses. These new results are presented in Supplemental Figure 3 and indicate that anxiety-related behavior significantly correlate with several electrophysiological properties, while n-3 supplementation tends to abolish most correlations. CSDS does not seem to modulate these correlations.
Some concerns should be addressed properly in order to be ready for publication.
Comments
1) The authors need to investigate body weight of mice and the amounts of diet consumed. These data are important to estimate how much fatty acids the mice consumed.
We would like to apologize for not having put these important data in the manuscript. Animal body weights during the procedure are now provided (see Supplemental Figure 4A). As described, stress had a significant effect on body weight, expressed in both g and % of day 1 (three-way ANOVA, Supplemental Table 1). Furthermore, n-3 PUFAs dietary supplementation also induced a significant effect on mice body weight. There were also significant interactions between stress and diet in our experiment.
Finally, we could not gather data on individual food intake, as animals are housed in groups of 2 per cage to undergo CSDS. However, we have estimated an average food consumption over the experimental procedure (Supplemental Figure 4B), which was calculated as the average daily food intake during the procedure per cage, divided by the number of animals in each cage (n=2). These results indicate that food intake is not altered by n-3 dietary supplementation. Unfortunately, we did not have access to data related to food intake in mice undergoing CSDS.
2) The authors found increased anxiety-related behaviors in the open field and light-dark tests but not in the elevated plus maze test. It would be better to discuss why such inconsistent behavioral consequences were observed between the behavioral tests. Especially, I am wondering the reason they performed the open field and light-dark tests under relatively bright environment (at 300 lux) but the elevated plus maze test under dim environment (at 15 lux). Could such difference in the floor brightness be account for aforementioned behavioral inconsistency?
We used different lighting intensity in the behavioral tests, as each test requires a specific light intensity to measure behavioral outcome. Thus, when using different behavioral tests, different light intensities are chosen. Here, we used previously-published protocols, using 300 Lux in the open field test (PMID 30937403), 15 Lux in the elevated plus maze (PMID 31075295), 300 Lux in the light/dark test (PMID 32317926) and 30 Lux during the social interaction tests (PMID 32317926).
3) Did the exposure to the EPA- and DHA-rich diet increase the social interaction ratio in naive mice? This would be useful to consider whether the exposure to the EPA- and DHA-rich diet increases the stress-resilience mice or enhances sociability of mice.
The reviewer is raising an interesting question. We have analyzed the social interaction ratio of naïve mice (absence of CSDS) in the two diet groups (control and n-3 suppl). Please see the updated Figure 5A, showing no effect of the diet on the social interaction ratio of naïve mice. The corresponding statistic, using a two-way ANOVA, has been described in the text and is also included in Supplemental Table 1. Thus, we can conclude that EPA-DHA rich diets do not increase social interaction nor sociability.
4) The authors need to justify that only males were used in this study or discuss this limitation.
Male mice were used throughout due to the procedure required for the induction of CSDS. Indeed, since aggressor mice are old CD1 Swiss retired male breeders (due to their increased aggressively), female C57Bl6/j mice cannot be used, as this would rather produce mating instead of aggression. While we acknowledge the fact that CSDS protocols have been adapted to perform these experiments with female mice (please see the following papers PMID: 29090682, 34587653, 28993631 and 28888327), this was outside of the scope of our study, as different paradigm would have been required. This point is now discussed at the end of the discussion.
Reviewer 2 Report
This study by Mathieu and colleagues is fascinating that long-chain n-3 polyunsaturated fatty acid is essential for neuronal transmission associated with normal behavior, which may play a role in the underlying molecular mechanisms for modulating anxiety and MDD. I have a few concerns about this study:
1. The data presentation is particularly confusing; the statistics are in Table S1; I think the degrees of statistical significance should be listed on the results so that readers will easily identify which result is significantly different.
2. Table 1 is confusing to read. It should be presented in a graph showing the statistical significance of asterisks.
3. Table S1 is challenging to read. It has multiple sections. The authors need to refer to each section specifically when mentioning them in the text.
4. Each Result section is loaded with information; it would help the readers understand the central message of each result if the authors added a conclusion sentence at the end of each Result section.
5. Figure 2 E does not have a key to describe the color code.
6. Baseline EPSC should be compared between treatment groups by Input-output curves (input - increasing strength of stimulation intensity; output-EPSC amplitude). This will resolve the EPSC magnitudes based on the sensitivity of neurons to different strengths of stimulation intensity. Only an input-output curve would best demonstrate the difference in baseline EPSC.
7. Given that LTD was not different between defeat- and defeat+ mice regardless of the n-3 suppl, and also there were some changes in the EPM behavior that suggest possible altered long-term performance, I think assessing the Long-term potentiation (LTP) in the medium spiny neurons will help explain the observed subtle behavioral deficits. Importantly, it may provide insights into how n-3 supplementation may improve LTP.
Author Response
Reviewer 2
This study by Mathieu and colleagues is fascinating that long-chain n-3 polyunsaturated fatty acid is essential for neuronal transmission associated with normal behavior, which may play a role in the underlying molecular mechanisms for modulating anxiety and MDD. I have a few concerns about this study:
We would like to thank both reviewers for their insightful and positive reviews of our manuscript. We highly appreciate the inputs which, we believe, significantly improved the quality of our manuscript. You will find below our responses to the questions raised. The changes have been highlighted in yellow in the updated version of the manuscript.
We would like to thank this reviewer for providing positive feedback on our work.
- The data presentation is particularly confusing; the statistics are in Table S1; I think the degrees of statistical significance should be listed on the results so that readers will easily identify which result is significantly different.
We thank the reviewer for pointing this out. This has been amended throughout. The statistical results are now included within the result narrative. We also provide in the current version an updated statistical summary (Supplemental Table 1), which is similar to the one published in a previous study from our laboratory (please see PMID: 32142670).
- Table 1 is confusing to read. It should be presented in a graph showing the statistical significance of asterisks.
Thank you for your suggestion, the data is now in the form of a graph. Please see the new Figure 2. Please also note that due to the addition of such a Figure within the manuscript, the numbering of subsequent figures has been incremented by 1.
- Table S1 is challenging to read. It has multiple sections. The authors need to refer to each section specifically when mentioning them in the text.
This has now been performed, throughout.
- Each Result section is loaded with information; it would help the readers understand the central message of each result if the authors added a conclusion sentence at the end of each Result section.
This has now been performed, throughout.
- Figure 2 E does not have a key to describe the color code.
We apologize, this has now been amended.
- Baseline EPSC should be compared between treatment groups by Input-output curves (input - increasing strength of stimulation intensity; output-EPSC amplitude). This will resolve the EPSC magnitudes based on the sensitivity of neurons to different strengths of stimulation intensity. Only an input-output curve would best demonstrate the difference in baseline EPSC.
We thank the reviewer for such an insight. We have now performed such an analysis. Please see Supplemental Figure 2 for data representation and Supplemental Table 1 for the corresponding statistics (three-way repeated-measures ANOVA). The result section has been adapted accordingly, and is now mentioning these new results.
- Given that LTD was not different between defeat- and defeat+ mice regardless of the n-3 suppl, and also there were some changes in the EPM behavior that suggest possible altered long-term performance, I think assessing the Long-term potentiation (LTP) in the medium spiny neurons will help explain the observed subtle behavioral deficits. Importantly, it may provide insights into how n-3 supplementation may improve LTP.
We thank the reviewer for these suggestions. LTP was not assessed in our current study, as we only focused on LTD, according to what our team has previously demonstrated (PMID: 27452462). However, we acknowledge that this might be an important limitation. Thus, we now discuss the following at the end of the discussion:
(i) a previous study linking DHA and hippocampal LTD.
(ii) a previous study demonstrating that maternal n-3 PUFAs dietary supplementation can promote hippocampal LTP in pups.
(iii) impairments of hippocampal LTP following CSDS in mice and rats.
(iv) the void in the literature regarding accumbal LTP, CSDS and n-3 PUFAs dietary supplementation.
Round 2
Reviewer 1 Report
The authors have addressed my previous concerns.